

# The neural machine translation models for the low-resource Kazakh–English language pair

Vladislav Karyukin[1], Diana Rakhimova[1,2], Aidana Karibayeva[1], Aliya Turganbayeva[1] and Asem Turarbek[1]

[1] Department of Information Systems, Al-Farabi Kazakh National University, Almaty, Kazakhstan
[2] Institute of Information and Computational Technologies, Almaty, Kazakhstan

## ABSTRACT

The development of the machine translation field was driven by people's need to communicate with each other globally by automatically translating words, sentences, and texts from one language into another. The neural machine translation approach has become one of the most significant in recent years. This approach requires large parallel corpora not available for low-resource languages, such as the Kazakh language, which makes it difficult to achieve the high performance of the neural machine translation models. This article explores the existing methods for dealing with low-resource languages by artificially increasing the size of the corpora and improving the performance of the Kazakh–English machine translation models. These methods are called forward translation, backward translation, and transfer learning. Then the Sequence-to-Sequence (recurrent neural network and bidirectional recurrent neural network) and Transformer neural machine translation architectures with their features and specifications are concerned for conducting experiments in training models on parallel corpora. The experimental part focuses on building translation models for the high-quality translation of formal social, political, and scientific texts with the synthetic parallel sentences from existing monolingual data in the Kazakh language using the forward translation approach and combining them with the parallel corpora parsed from the official government websites. The total corpora of 380,000 parallel Kazakh–English sentences are trained on the recurrent neural network, bidirectional recurrent neural network, and Transformer models of the OpenNMT framework. The quality of the trained model is evaluated with the BLEU, WER, and TER metrics. Moreover, the sample translations were also analyzed. The RNN and BRNN models showed a more precise translation than the Transformer model. The Byte-Pair Encoding tokenization technique showed better metrics scores and translation than the word tokenization technique. The Bidirectional recurrent neural network with the Byte-Pair Encoding technique showed the best performance with 0.49 BLEU, 0.51 WER, and 0.45 TER.

Corresponding author
Vladislav Karyukin,
vladislav.karyukin@gmail.com

# INTRODUCTION

Machine translation (MT) (*Mohamed et al., 2021*) is a field of natural language processing (NLP) that defines the translation of words, sentences, paragraphs, and whole texts from one language into another. The language barrier remains a serious problem today. Many different approaches, including rule-based (*Islam, Anik & Al Islam, 2021*) and statistical (*Babhulgaonkar & Bharad, 2017*), were used to reach a high level of MT in a large variety of language pairs. Although these approaches were effective, they were mostly completely replaced by the rapidly developing neural machine translation (NMT) (*Singh et al., 2019*) method that uses huge parallel corpora. *Zhao, Gao & Fang (2021)* used the Google Transformer model to translate English texts into Chinese. The model was trained on English–Chinese (EN–CH) sentences with one or more GPUs. In the experimental results, the BLEU score of 0.29 was achieved. Three RNN-based NMT models were used for the Chinese–English and German–English MT tasks. The achieved BLEU score values were 0.36–0.37.

Many high-resource languages, such as English, German, Spanish, French, Italian, Chinese, *etc.*, do not have problems acquiring large parallel datasets as researchers have created huge amounts of corpora in recent years. It is freely available online, especially on such websites as OPUS (*Tiedemann, 2022*), WMT (*Koehn et al., 2022*), and Paracrawl (*Van der Linde, 2022*). It is possible to download more than 300 million sentence pairs for English–German (EN–DE), 400 million for English–Spanish (EN–ES), 350 million for English–French (EN–FR), and 150 million for English–Italian (EN–IT).

Moreover, the variety of topics covered by these language pairs is wide. Therefore, it is not urgent to search for other monolingual and bilingual data sources for these languages. Instead, it is possible to focus on training NMT models and compare their results. Despite an impressive number of parallel corpora for the language pairs mentioned above, the situation is not so impressive for other languages, so-called low-resource ones (*Jooste, Haque & Way, 2022*), representing languages significantly underrepresented online.

The necessity of the fast development of MT for low-resource languages ignited the appearance of a large number of works in this field. Consequently, NMT occupies an important place among the works devoted to MT. The problem of the low-resource neural MT for the Spanish–Farsi pair of languages was explored in *Ahmadnia, Aranovich & Dorr (2021)*, where 0.5 M sentences from the Opensubtitles2018 corpora were combined with 50,000 sentences from the Tanzil collected. The Transformer model on top of the PyTorch framework was utilized for training an NMT model. The best values of BLEU and TER scores achieved 0.39 and 0.47, correspondingly. Another NMT experiment for the low-resource English–Irish pair of languages with the RNN and Transformer models with Byte Pair Encoding (BPE) (*Nonaka et al., 2022*) and unigram approaches was conducted in *Lankford, Afli & Way (2022)*. The values of the BLEU and TER reached 0.56 and 0.39 for the RNN model and 0.59 and 0.35 for the Transformer model. *Kandimalla et al. (2022)* investigated NMT for the English–Hindi and English–Bengali pairs of languages using the Transformer models from the OpenNMT tool. The BLEU score of 0.39 was achieved for English–Hindi and 0.23 for English–Bengali. *Góngora, Giossa & Chiruzzo (2022)* explored

the MT problems for the low-resource Guarani–Spanish language pair using pre-trained word embeddings. The MT models trained on 383 collected parallel sentences got the BLEU score from 0.15 to 0.22.

In *Ngo et al. (2022)*, synthetic Chinese-Vietnamese and Japanese-Vietnamese parallel corpora were formed, and NMT models trained on them reached 0.17 and 0.18 BLEU scores.

The problems of the low-resource Asian languages were explored in *Rubino et al. (2020)*. Eight translation directions from the English language to one of the Asian languages were described here with the formation of initial monolingual corpora to the bilingual one and combination with the existing parallel sentences. A strong baseline Transformer NMT architecture heavily relying on a multi-head attention mechanism was utilized for training models for each pair of languages. The initial values of the BLEU scores were very low, but after hyper-parameter tuning, the values reached the range of 0.22 to 0.33. The problem of a small number of available data for the low-resource languages was also inspected in *Abdulmumin et al. (2020)*. This work proposed a novel approach to both the backward and forward translation models to benefit from the monolingual target data. The experiments were conducted on the English–German corpora. After the last eight checkpoints of the training steps, the average BLEU score reached 0.21.

*Edunov et al. (2018)* augmented the parallel corpora with the texts generated by a back-translation approach to improving NMT. The experimental results allowed achieving the BLEU scores of 0.35 and 0.46, respectively. *Sennrich, Haddow & Birch (2016)* also explored the possibility of training with monolingual data by using the back-translation approach and pairing synthetic data with existing parallel corpora, substantially improving the BLEU score from 0.18 to 0.20. *Ha, Niehues & Waibel (2016)* extended the multilingual scenario of NMT, including low-resource languages, which improved MT by 0.26. The experiments integrated the encoder-decoder architecture.

The Kazakh language also belongs to the category of low-resource languages. Therefore, it is challenging not only to find high-quality parallel corpora for the Kazakh–English (KZ–EN) language pair (*Rakhimova et al., 2021*), but even the monolingual texts on the Internet are not represented well. For example, on the OPUS website, there are only about 1 million parallel sentences for this language pair. In addition, the manual analysis of the sentences from that source shows that the topics of the sentences are seriously restricted, and the quality of sentences is shallow. In this way, relying on this kind of data is tough for any serious MT field experiments. Furthermore, the number of websites in the Kazakh language is also restricted, and some web pages are not fully translated into Kazakh, leaving many blank spaces. Thus, government websites, news portals, books, and scientific articles have become more trusted sources of high-quality texts in the Kazakh language.

The works devoted to NMT of the Turkic language group and the Kazakh language are not widely presented. Nevertheless, *Khusainov et al. (2018)* assessed the possibility of combining the rule-based and the neural network approaches to constructing the MT system for the Tatar-Russian language pair. The Nematus toolkit with default hyperparameters values was used to train on the newly collected corpora of 0.5 M sentence

pairs. In every test set, 1,000 sentences were randomly selected. The values of the BLEU score were in the range of 0.29 to 0.63. The study (*Zhanabergenova & Tukeyev, 2021*; *Tukeyev et al., 2020*) introduced a morphological segmentation for the Kazakh language based on the complete set of endings (CSE). *Tukeyev, Karibayeva & Abduali (2018)* used the Tensorflow Seq2Seq model in the experimental part for training the parallel corpora of the KZ–EN 109,772 sentences, resulting in the BLEU score values from 0.18 to 0.25. *Toral et al. (2019)* used the available KZ–EN corpora from the WMT19 and some portion of synthetic data to train the NMT models. The BLEU score with the BPE segmentation reached the values of 0.23 and 0.22. In *Turganbayeva et al. (2022)*, the grammatical structure of complex Kazakh sentences was observed. This study proved the methods used worked; however, they completely relied on the rule-based approach. *Niyazbek, Talp & Sun (2021)* used the rule-based dictionary method for Chinese–Kazakh (CH–KZ) MT. So many research works related to MT of the Kazakh language are restricted by the rule-based method leaving the NMT approach open to development. Moreover, the ways of the corpora formation are also not thoroughly described.

This article discusses the methods for generating high-quality synthetic corpora to train MT models for the KZ–EN pair by choosing the most effective techniques for the low-resource languages. Using the most effective architectures in NMT, such as Sequence-to-Sequence (recurrent neural network and bidirectional recurrent neural network) and Transformer, the experimental results strive to achieve high values of BLEU, WER, and TER scores while building a translator that can provide, first of all, the high-quality translation of formal social, political, and scientific texts.

For ease of the content of the article understanding, Table 1 includes the list of abbreviations.

The rest of the article is organized in the following way: The synthetic corpora formation approaches and transfer learning section describes backward translation, forward translation, and transfer learning approaches for dealing with low-resource languages. The Materials & Methods section describes corpora formation steps, the Sequence-to-Sequence (Seq2Seq), recurrent neural network (RNN), bidirectional recurrent neural network (BRNN), and Transformer NMT architectures for parallel training corpora, tokenization techniques, and model evaluation metrics. The Experiments and Results section describes the experiments for generating KZ–EN parallel corpora and creating the MT model. Finally, the Conclusions and Future Works section summarizes the previous parts of the article and proposes the groundwork for future research.

## Synthetic corpora formation approaches and transfer learning

When dealing with low-resource languages, it is important to use approaches that increase the corpora size to reach a higher quality of MT. There are generally three approaches to enhancing the size of the corpora and the performance of the models: forward translation (FT) (*Zhang & Zong, 2020*), backward translation (BT) (*Abdulmumin, Galadanci & Isa, 2020*), and transfer learning (TL) (*Wu et al., 2022*).

BT is generally a quality control method for translations. In this method, the monolingual data of the target language is translated back to its source language by an

**Table 1** The list of abbreviations.

| No | Abbreviation | Explanation |
|----|--------------|-------------|
| 1 | MT | Machine Translation |
| 2 | NMT | Neural Machine Translation |
| 3 | NLP | Natural Language Processing |
| 4 | WMT | Workshop on Machine Translation |
| 5 | Seq2Seq | Sequence-to-Sequence |
| 6 | RNN | Recurrent Neural Network |
| 7 | BRNN | Bidirectional Recurrent Neural Network |
| 8 | GPU | Graphics Processing Unit |
| 9 | BLEU | Bilingual Evaluation Understudy |
| 10 | WER | Word Error Rate |
| 11 | TER | Translation Error Rate |
| 12 | FT | Forward Translation |
| 13 | BT | Backward Translation |
| 14 | TL | Transfer Learning |
| 15 | KZ–EN | Kazakh-English |
| 16 | CH–KZ | Chinese–Kazakh |
| 17 | EN–DE | English–German |
| 18 | EN–ES | English–Spanish |
| 19 | EN–FR | English–French |
| 20 | EN–IT | English–Italian |

independent translator or a translation system. Then the generated synthetic data is added to the existing parallel corpora to increase their size. BT is also used to check the quality of the original parallel corpora. This way, the translated texts are checked by comparing them with the original variant. It can help identify errors, ambiguity, or confusion that may arise with the translation. However, the synthetic data received by BT is usually much noisier than the original data, and due to the nature of the language, a 100% perfect match is generally unattainable by this method. Since the goal of translation is to preserve the meaning of the translated text, two translators can offer two different translations that retain the original text's meaning. BT is a three-stage translation quality control method that includes the translation of the completed translation back into the original language, the comparison of this new translation with the original text, and the reconciliation of any material differences between them.

Generally, the BT approach is implemented by the following series of steps:

– There is a translation platform–$T_p$ and the target data–$Y = \{y\}_{i=1}^{N}$, where $y$ is a target text and $N$ is the total number of texts;

– The $T_p$ is used on the target data $Y$ to get the source data $X = \{x\}_{i=1}^{N}$, where $x$ is a source text and $N$ is the total number of texts;

– The newly generated parallel corpora $D^*$ is received by combining $X$ and $Y$;

– Then the initially existing parallel corpora $D$ are added to $D^*$.

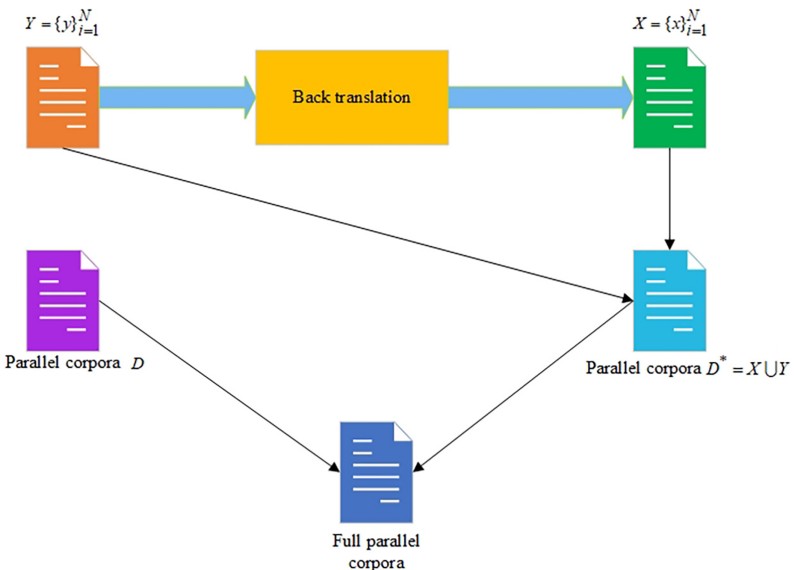

**Figure 1** **The backward translation approach for increasing the size of the parallel corpora.**

The scheme of the BT approach is shown in Fig. 1.

FT is the opposite approach to BT, where the monolingual data of the source language is translated to the target language by the translation system. While this method allows synthesizing parallel corpora, it is also used for quality control of the existing parallel corpora in the same way BT did.

The FT approach is implemented by the following series of steps:

– There is a translation platform–$T_p$ and the source data–$X = \{x\}_{i=1}^{N}$ where $x$ is a source text and $N$ is the total number of texts.

– The $T_p$ is used on the source data $X$ to get the target data $Y = \{y\}_{i=1}^{N}$, where $y$ is a target text and $N$ is the total number of texts.

– The newly generated parallel corpora $D^*$ are received by combining $X$ and $Y$.

– Then the initially existing parallel corpora $D$ are added to $D^*$.

The scheme of the FT approach is shown in Fig. 2.

Another technique commonly used in the low-resource MT scenario is TL. This approach uses a high-resource language pair to train the model, which is then applied to the low-resource pair. The proposed method enhances the performance of a low-resource model trained from scratch. Here we have $L_1 \rightarrow L_3$ as a high-resource language pair and $L_2 \rightarrow L_3$ as a low-resource pair. $L_1$ and $L_2$ are source languages, and $L_3$ is a target language for both pairs. $M_{L_1 \rightarrow L_3}$ is a parent model trained on a $L_1 \rightarrow L_3$ pair. The parameters of this model are used to fine-tune a $M_{L_2 \rightarrow L_3}$ model that uses a $L_2 \rightarrow L_3$ pair. The scheme of the TL technique is shown in Fig. 3.

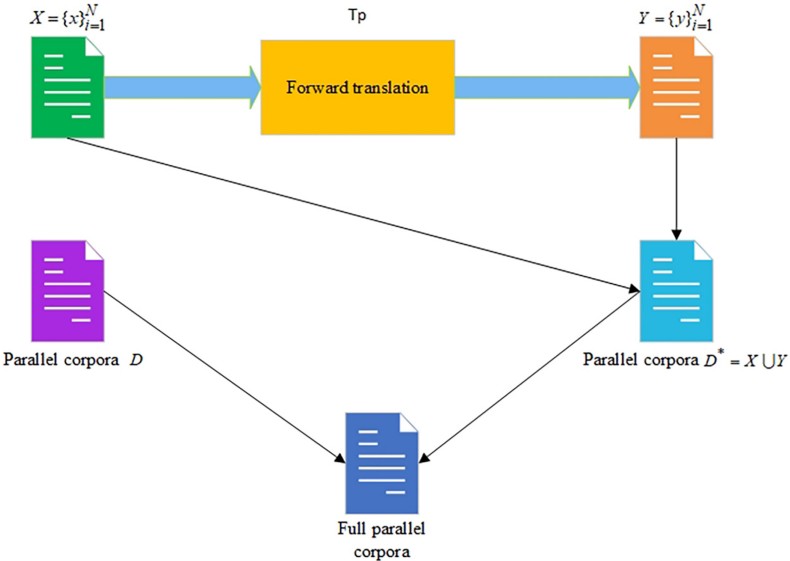

**Figure 2 The forward translation approach for increasing the size of the parallel corpora.**

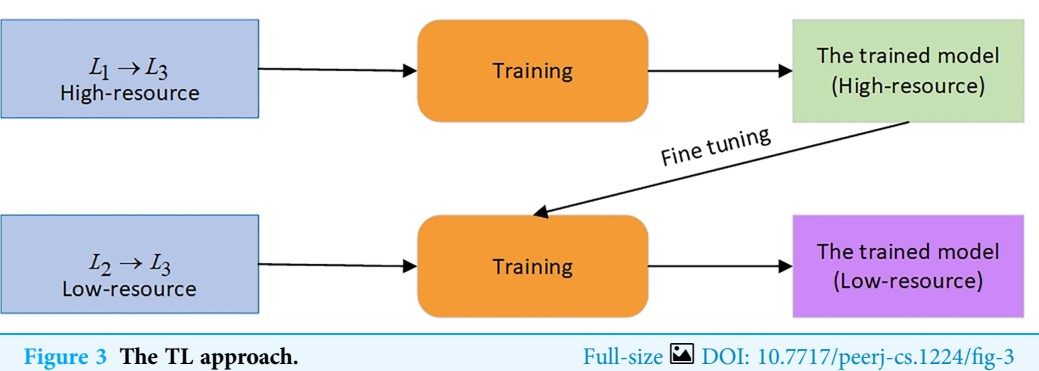

**Figure 3 The TL approach.**

The presented methods are directed at solving the problem of low-resource languages. In addition to dealing with the low-resource problem, the NMT architectures are also very important in increasing the performance of the trained models.

# MATERIALS AND METHODS

## Corpora formation

The initial KZ–EN parallel corpora were parsed from Kazakhstan's official bilingual government Internet sources (*The Republic of Kazakhstan, 2023*; *Strategy, 2022*; *Primeminister, 2022*) in the amount of 205,000 sentences with the use of the Bitextor tool. The crawled sentences were preprocessed (normalized, cleaned from extra symbols, characters, and empty strings, and aligned). Then the monolingual texts in the Kazakh language were gained from the scientific articles published in the Kazakh language in scientific journals (*Kalekeyeva & Litvinov, 2021*; *Sapakova, 2022*) that were taken because they had high-quality texts edited by the authors and checked by the editors. The texts from the scientific articles were also preprocessed (irrelevant symbols, characters, and

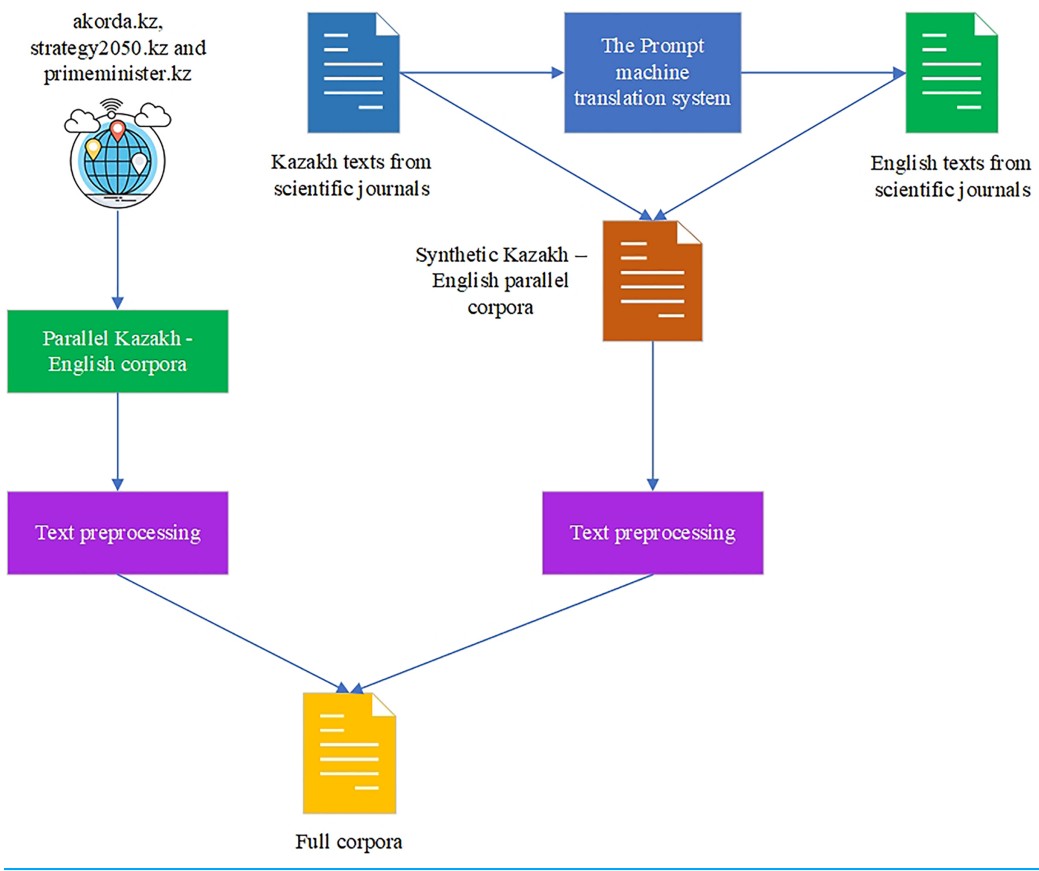

**Figure 4 Full corpora formation.**

words were deleted, and empty sentences were removed). The available open-source parallel corpora from the OPUS website were not considered for training because they included very restricted thematic unrelated to political or scientific topics, and the quality of texts was generally poor. The monolingual Kazakh texts in the amount of 205,000 sentences from scientific journals were translated with the use of the Promt MT system (the FT approach was applied). This platform supports different pairs of languages and is good for the KZ–EN translation.

After the crawled and synthetically generated KZ–EN corpora are gained, they are combined together to form the full corpora of 380,000 parallel sentences. The corpora are available on GitHub (*Karyukin, 2022*). The scheme of the parallel corpora formation is shown in Fig. 4.

The combined corpora went again through the quality-checking phase, where the sentences containing only mathematical symbols and other characters and sentences with poor translation were removed.

## The NMT architectures

The fully formed KZ–EN parallel corpora are used for training NMT models. The NMT architectures have been intensively developing for the last decade. The traditionally used NMT architecture was Seq2Seq (*Sindhu, Guha & Singh Panwar, 2022*) for translating from

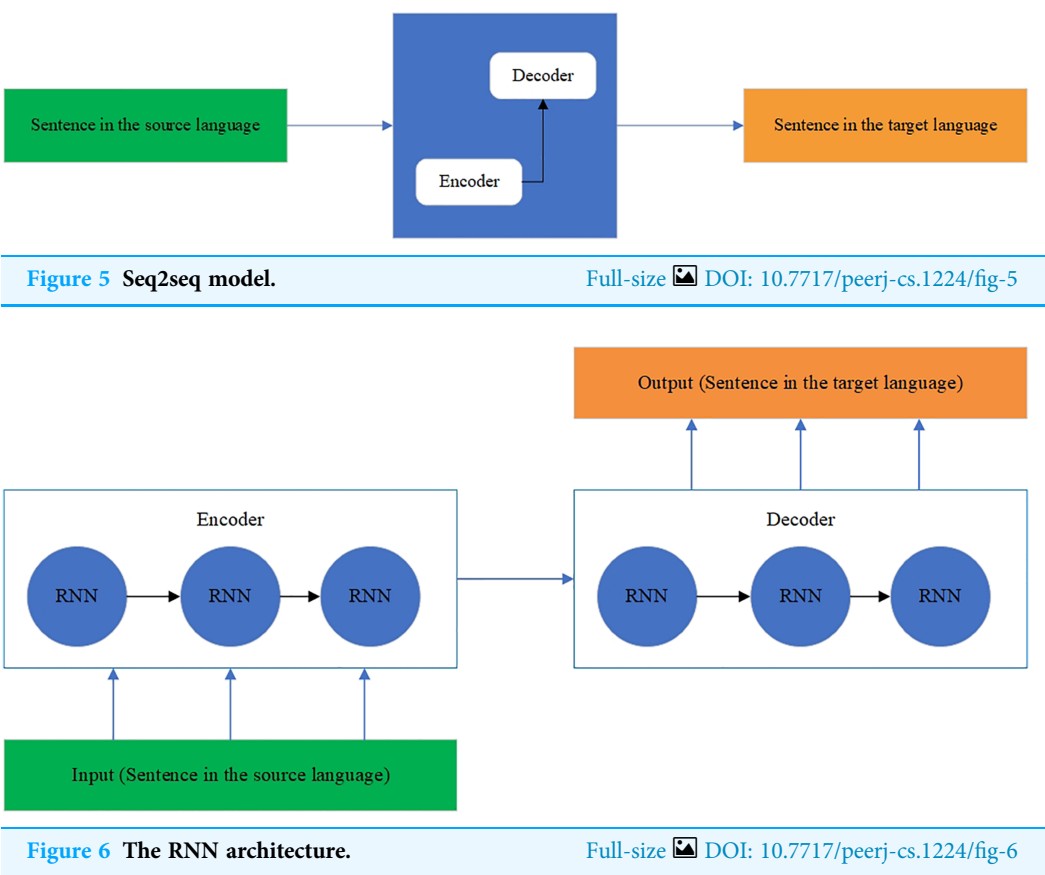

**Figure 5  Seq2seq model.**     

**Figure 6  The RNN architecture.**     

one language to another. It is a deep learning model that takes a sequence of elements (words, letters, time series, *etc.*) and outputs another sequence of elements. The model consists of two main components: an encoder and a decoder. First, the encoder processes all inputs by transforming them into a single vector called a context. The context contains all the information that the encoder discovers from the input. Then, the vector is sent to the decoder, which creates an output sequence. Because the task is sequence-based, both the encoder and decoder tend to use some form of RNN, such as LSTM, GRU, *etc.* The hidden state vector can be of any size, although in most cases, it is taken as a power of two (layer) and length (128, 256, 512, 1,024), which can somehow reflect the complexity of the entire sequence. The scheme of the seq2seq model is shown in Fig. 5.

The RNN encoder and decoder model (*Sharma et al., 2021*) is designed to process a sequence of input texts and give a sequence of output texts. The hidden states contain loops where output at a one-time step becomes an input at another time step, defining a memory form. Therefore, the whole architecture of an RNN includes the following elements:

– Inputs are words of a sentence in one language.
– Embedding layers convert all words to corresponding vectors.
– The encoder applies the context of word vectors from a previous time step to the current word vector.
– The decoder transforms the encoded input into the translation sequence.

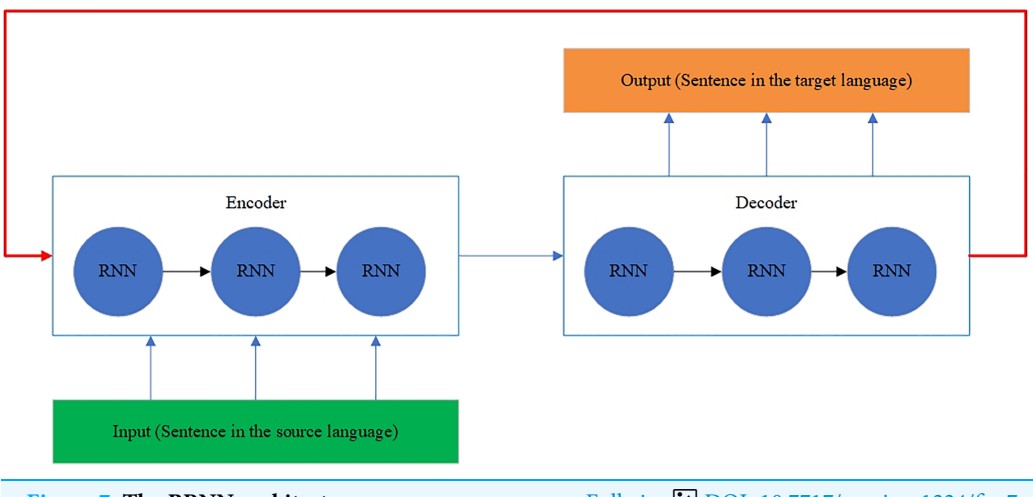

**Figure 7** **The BRNN architecture.**

– Output is a translated sentence.

The RNN architecture is shown in Fig. 6 (*Sharma et al., 2021*). The RNN architecture is also modified to model the history and future generated context (*Shanmugavadivel et al., 2022*). The neurons are split into the parts responsible for the forward and backward directions. At the same time, outputs from the forward states are not connected to inputs of the backward states, and it is also applied to the reverse as well. This designed architecture transforms into a unidirectional RNN without the backward states. The BRNN architecture is shown in Fig. 7 (*Shanmugavadivel et al., 2022*).

Recently, a new MT architecture called the Transformer model (*Wan et al., 2022*) was introduced. In this implementation, the RNN layers are replaced with the attention ones. As a result, the Transformer does not process input text sentences in sequential order. Instead, the self-attention mechanism identifies the context that gives meaning to an input sequence. Therefore, it provides more parallelization and reduces the training time.

The Transformer architecture is shown in Fig. 8 (*Wan et al., 2022*).

## Tokenization techniques

The texts of the parallel corpora go through tokenization before being trained by NMT models. In NLP, tokenization is dividing the raw text into smaller units called tokens. The widely used tokenization techniques are word, character, and Byte-Pair Encoding (BPE).

In word tokenization, sentences are split by space into words, and these words are considered to be tokens. For example, the sentence "This model will be trained" is tokenized by words as {'This', 'model', 'will', 'be', 'trained'}. Although this technique is simple and clear, it requires to have a huge vocabulary to train a good MT model, or the words that are not present in it will be translated as <UNK>.

The character tokenization technique is designed to solve the problem of a huge vocabulary for word tokenization. In this technique, the word is split by each character. For example, "Model" is tokenized as {'M', 'o', 'd', 'e', 'l'}. The disadvantage of this tokenization technique is the requirement of large computational resources to train an MT model.
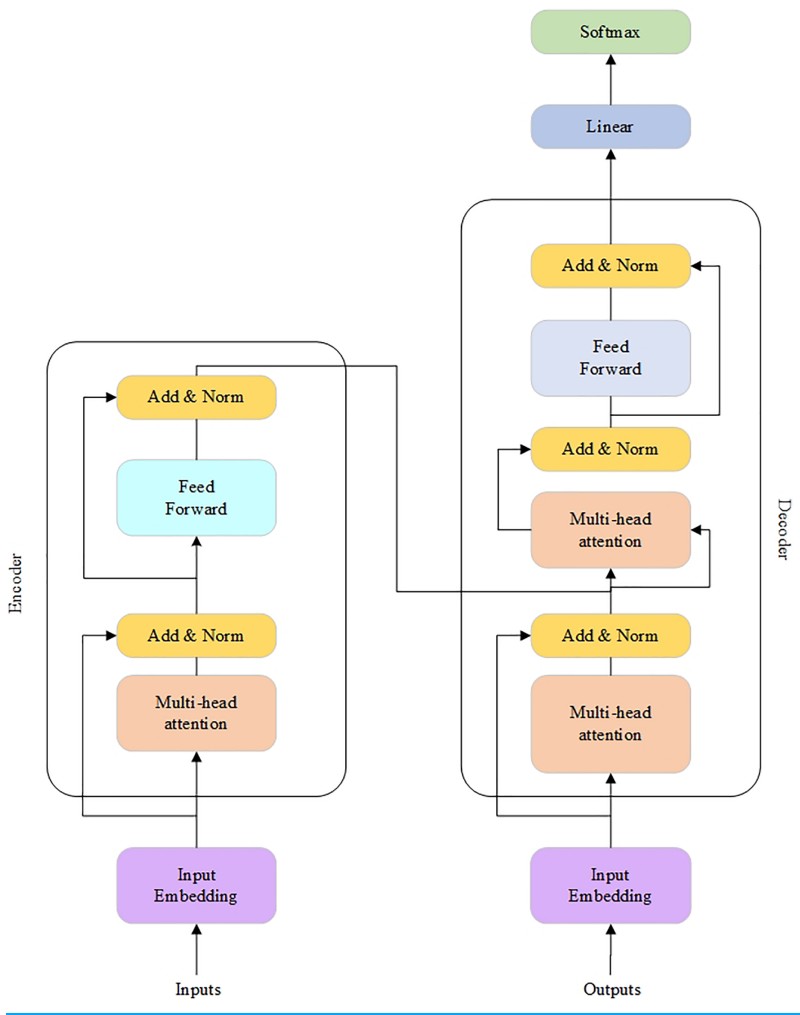

**Figure 8** The Transformer architecture.

   Subword tokenization is a technique between word and character tokenization methods. It splits words into smaller subwords. For example, "Models" is split into "Model" and "s". "Training" is split into "Train" and "ing". One of the most popular subword tokenization algorithms is Byte-Pair Encoding (BPE), where the data compression algorithm is utilized.

## Model evaluation

The quality of MT models is evaluated with the BLEU (*Mouratidis, Kermanidis & Sosoni, 2020*), WER (*Stanojević et al., 2015*), and TER (*Bojar, Graham & Kamran, 2017*) metrics, which show a high correlation with human quality ratings. These metrics' values range from 0 to 1 (from 0% to 100% if the percentage measure is used). The BLEU metric is based on the idea that "the closer the MT to the professional human translation, the better it is."

It is computed as the ratio of the number of words of the translation candidate found in the reference translation to the total number of words of the candidate:

$$BLEU = Brevity\_Penalty \times \prod_{n=1}^{N} p_n^{w_n}, \tag{1}$$

where $p_n$ are the precision values of the N-gram; N is the maximum level of granularity (unigram, bigram, trigram, *etc.*); $w_n$ are the weights of the N-gram; Brevity_Penalty is the measure that penalizes sentences that are too short:

$$Brevity\_Penalty = \begin{cases} 1, c > r \\ e^{(1-r/c)}, c \leq r \end{cases}, \tag{2}$$

where $c$ is the length of the candidate translation and $r$ is the length of the reference.

The closer the BLEU score value to 1 (100%), the higher the translation quality is.

The WER metric finds the distance between a translation candidate and its reference. The minimum number of edit operations (insertion, substitution, and deletion) is divided by the number of words in the reference:

$$WER(c, r) = \frac{\min(e)}{r}, \tag{3}$$

where $r$ is the reference; $c$ is the candidate; $e$ is the number of edit operations. The quality of the WER metric is evaluated opposite to the BLEU metric. The closer the WER score value to 0(0%), the better the translation is.

The WER metric has a significant drawback because it does not consider shifts of words. The TER metric improves this case by considering shifts in sentences. Thus, it is calculated as follows:

$$TER(c, r) = \frac{\min(e + shifts)}{r}, \tag{4}$$

where $r$ is the reference; $c$ is the candidate; $e$ is the number of edit operations; *shifts* is the number of shift operations. The quality of the TER metric is evaluated similarly to the WER metric. The closer the TER score value to 0-(0%), the better the translation is.

## EXPERIMENTS AND RESULTS

### Training MT models

The formed full parallel KZ–EN corpora were moved to the training phase with the described NMT models. The OpenNMT framework, an open-source NMT application, was utilized for this purpose. This tool contains many different architectures whose parameters are easy to configure. The RNN, BRNN, and Transformer models were tuned to train MT models. The sentences were tokenized with word and BPE subword tokenization techniques. The character tokenization technique was not implemented because it used large computational resources. The RNN, BRNN, and Transformer MT models had the following parameters for the training phase (Table 2). The full corpora were divided into 90% for training, 5% for validating, and 5% for testing phases, leaving the

**Table 2 The parameters of the neural machine translation models.**

| RNN | BRNN | Transformer |
|---|---|---|
| #Model<br>transforms: onmt_tokenize<br>encoder_type: rnn<br>decoder_type: rnn<br>rnn_type: LSTM | #Model<br>transforms: onmt_tokenize<br>encoder_type: brnn<br>decoder_type: rnn | #Model<br>encoder_type: transformer<br>decoder_type: transformer<br>position_encoding: true<br>enc_layers: 6<br>dec_layers: 6<br>heads: 8<br>rnn_size: 512<br>word_vec_size: 512<br>transformer_ff: 2048<br>dropout_steps: [0]<br>dropout: [0.1]<br>attention_dropout: [0.1] |
| #Training steps and stopping criteria<br>train_steps: 100000<br>valid_steps: 10000<br>early_stopping: 4<br>early_stopping_criteria: accuracy<br>early_stopping_criteria: ppl | #Training steps and stopping criteria<br>train_steps: 100000<br>valid_steps: 10000<br>early_stopping: 4<br>early_stopping_criteria: accuracy<br>early_stopping_criteria: ppl | #Training steps and stopping criteria<br>train_steps: 100000<br>valid_steps: 5000<br>early_stopping: 4<br>early_stopping_criteria: accuracy<br>early_stopping_criteria: ppl |
| #Infrastructure<br>world_size: 1<br>gpu_ranks: [0] | #Infrastructure<br>world_size: 1<br>gpu_ranks: [0] | # Train on two GPUs<br>world_size: 2<br>gpu_ranks: [0, 1] |

**Table 3 The neural machine translation architectures and tokenization techniques.**

| Architecture | BLEU | WER | TER |
|---|---|---|---|
| RNN_word_tokenization | 0.45 | 0.55 | 0.48 |
| BRNN_word_tokenization | 0.43 | 0.58 | 0.57 |
| Transformer_word_tokenization | 0.37 | 0.62 | 0.55 |
| RNN_bpe | 0.46 | 0.54 | 0.48 |
| BRNN_bpe | 0.49 | 0.51 | 0.45 |
| Transformer_bpe | 0.42 | 0.58 | 0.51 |

largest part for training to maximize the performance of the models. The configuration of the OpenNMT framework is available on GitHub (*Karyukin, 2022*).

The experiments were conducted on the workstation with the following specifications: Core i7 4790K, 32 GB RAM, 1 TB SSD, and NVIDIA Geforce RTX 2070 Super. In addition, for training the Transformer models, the second GPU, NVIDIA Geforce GTX 1080, was added to the bundle with RTX 2070 Super to launch those heavy models. The values of each of these metrics for every architecture with word tokenization and BPE are shown in Table 3.

The models showed a good performance for the KZ–EN corpora, comparable with the high-resource trained models. The examples of the translation results of test sentences are shown in Table 4. Here, *Source* is the sentence in the Kazakh language; *Target* is the

**Table 4 The Kazakh–English test translation samples.**

**Sentence 1**

| | |
|---|---|
| Source | өндірушілердің ауыл шаруашылығы өнімдеріне бағасының өзгеруі |
| Target | changes in producer prices for agricultural products |
| Predict—RNN (word tokenization) | changes in prices for agricultural products |
| Predict—RNN (bpe) | changes in prices for agricultural products |
| Predict—BRNN (word tokenization) | changing prices for agricultural products |
| Predict—BRNN (bpe) | changes in prices for agricultural products |
| Predict—Transformer (word tokenization) | changes in producer prices for agricultural products |
| Predict—Transformer (bpe) | changes in producer prices for agricultural products |

**Sentence 2**

| | |
|---|---|
| Source | бұл дамыту үшін үлкен мүмкіндік береді және әлемде қандай жағдайлардын болып жатқандығын білуге септігін тигізеді. |
| Target | it offers great opportunities for development and helps to find out what situations are happening in the world. |
| Predict—RNN (word tokenization) | this will make a great opportunity to develop and what is happening in the world. |
| Predict—RNN (bpe) | this gives great opportunities for development and contributes to what conditions are what conditions are happening in the world. |
| Predict—BRNN (word tokenization) | this will give a great opportunity for the development of this and will help you know what the world is. |
| Predict—BRNN (bpe) | this gives a great opportunity to develop and know what conditions are in the world. |
| Predict—Transformer (word tokenization) | this will have great opportunities for the development and will contribute to what is happening in the world. |
| Predict—Transformer (bpe) | this will make it possible to develop and know what conditions are taking place in the world. |

**Sentence 3**

| | |
|---|---|
| Source | еске сала кетейік, астанада 7-8 қыркүйек күндері хі еуразиялық kazenergy форумы өтіп жатыр. |
| Target | Recall that in the capital on September 7-8, the XI Eurasian Forum of kazenergy is held. |
| Predict—RNN (word tokenization) | Recall that in the capital on September 7-8 the Eurasian kazenergy forum is taking place. |
| Predict—RNN (bpe) | Recall that in the capital on September 7-7, xi Eurasian Energy Forum is taking place. |
| Predict—BRNN (word tokenization) | Recall that in the capital in September 7-8 the XI Eurasian kazenergy forum is taking place in the capital. |
| Predict—BRNN (bpe) | Recall that in the capital on September 7-7, the XI Eurasian kazenergy forum is being held. |
| Predict—Transformer (word tokenization) | Recall that the XIII Eurasian Forum of kazenergy is held in the capital. |
| Predict—Transformer (bpe) | Recall, on September 7-8, the Eurasian Forum of xi is held in the capital. |

**Sentence 4**

| | |
|---|---|
| Source | сапа көшбасшысы номинациясында жамбыл облысының "қазфосфат" жшс үздік атанды. |
| Target | in the nomination "Quality Leader" the best was Kazphosphate LLP of Zhambyl region. |
| Predict—RNN (word tokenization) | in the nomination of the quality leader, Zhambyl region became the best атанды |
| Predict—RNN (bpe) | In the nomination of the leader of the quality leader of the Zhambyl region, the best became Kazphosphate LLP. |
| Predict—BRNN (word tokenization) | In the nomination quality leader the best was the үздік LLP of Zhambyl region. |
| Predict—BRNN (bpe) | In the nomination of the quality leader, the best became the best was Kazphosphat LLP. |
| Predict—Transformer (word tokenization) | in the nomination "Best Quality. of Zhambyl Region. |
| Predict—Transformer (bpe) | in the nomination "Best quality Leader of the Nation of Zhambyl region" became the best. |

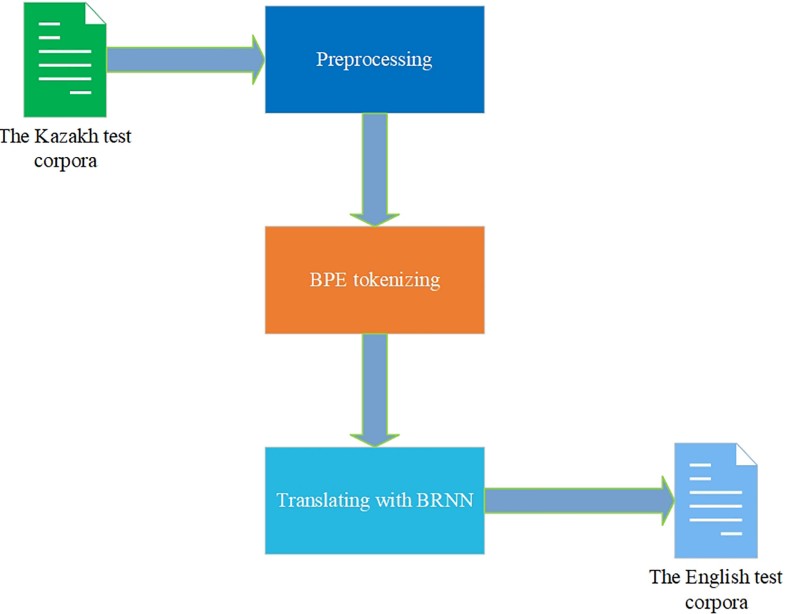

The Kazakh test corpora

Preprocessing

BPE tokenizing

Translating with BRNN

The English test corpora

**Figure 9  Translation with the trained model.**     

sentence in the English language; *Predict* is the translation of test sentences by the trained NMT models (RNN, BRNN, and Transformer).

Comparing the values of BLEU, WER, and TER metrics from Table 2 and translations from Table 3, it is seen that the RNN and BRNN models give a more precise translation than the Transformer model. The BPE tokenization technique showed better metrics scores and translation than the word tokenization technique. Nevertheless, analyzing the results of other works devoted to the NMT for the KZ–EN corpora, the improvements in the metrics scores are noticeable. Having the corpora mostly equal in size to the corpora of 400,000 parallel KZ–EN sentences in *Tukeyev, Karibayeva & Abduali (2018)* made it possible to achieve the best BLEU score of 0.16. The article (*Toral et al., 2019*) had only 67,000 parallel KZ–EN sentences, and the best BLEU score was 0.28. Despite the difference in the size and quality of corpora used in those works, the experimental results in this research show the importance of the corpora of parallel sentences and NMT architectures for achieving the best KZ–EN MT results. A very important note is that the RNN and BRNN models' scores were higher than the metrics' values of the Transformer model for the gathered corpora. The BPE tokenization technique outperformed the word tokenization for all the used models. All these NMT models can be used in the MT application currently under development by our research group. Now, the BRNN model with the BPE tokenization technique is adopted there (Fig. 9).

Although the evaluation metrics and translation samples in Tables 2 and 3 provide opportunities for comparing the results among sentences and different NMT models, it would also be preferable to have other Kazakh–English corpora with a number of sentences of around 500,000 to 1 million to get closer to the high-resource scenarios to have an even broader analysis. It is the task of future works.

## CONCLUSIONS AND FUTURE WORKS

This article explored building high-quality NMT models for the low-resource language pair, specifically KZ–EN. First, the article paid much attention to the size of the parallel corpora and the existing approaches for increasing their sizes for the low-resource languages, such as FT, BT, and TL. The features, characteristics, and schemes of the most popular MT architectures, Seq2Seq, RNN, BRNN, and Transformer, were also thoroughly described. Then the ways of the KZ–EN parallel corpora formation and the NMT models training methods were systematically described. Despite several hundred thousand parallel sentences of the language pairs with the Kazakh language, their quality and scarcity remain significant problems today. Moreover, those corpora do not fall within the scope of study of formal social, political, and scientific texts, which qualified translation this research was about to achieve. The corpora of 205,000 monolingual Kazakh sentences from the scientific articles were translated with the Promt MT system and combined with 175,000 parallel sentences parsed from the official government Internet sources, forming a very large *corpus* of 308,000 KZ–EN sentences. After the parallel corpora had been fully formed, the RNN, BRNN, and Transformer architectures of the OpenNMT framework were utilized for training advanced MT models. The quality of trained models was then evaluated with the BLEU, WER, and TER metrics, highlighting the best values of the RNN and BRRN architectures with the BPE technique. The BRNN model with the BPE tokenization technique is adopted for the new MT application which is under development. Although the gained parallel corpora and trained NMT model are very useful for MT of the Kazakh language, other tasks of great importance remain for future works. They include the following assignments:

–Increasing the size of the gathered corpora with more synthetic data.
–Parsing more texts from other multilingual websites.
–Trying the BT and TL techniques.
–Applying the synthetic corpora generation and NMT approaches for training other low-resource language pairs, *i.e.*, Kyrgyz–English, Uzbek–English.

### Funding

This research was performed and financed by the grant Project IRN AP 09259556 of the Ministry of Science and Higher Education of the Republic of Kazakhstan. The funders had no role in study design, data collection and analysis, decision to publish, or preparation of the manuscript.

### Grant Disclosures

The following grant information was disclosed by the authors:
Ministry of Science and Higher Education of the Republic of Kazakhstan: IRN AP: 09259556.

## Competing Interests

The authors declare that they have no competing interests.

## Author Contributions

- Vladislav Karyukin conceived and designed the experiments, performed the experiments, performed the computation work, prepared figures and/or tables, and approved the final draft.
- Diana Rakhimova conceived and designed the experiments, authored or reviewed drafts of the article, and approved the final draft.
- Aidana Karibayeva analyzed the data, authored or reviewed drafts of the article, and approved the final draft.
- Aliya Turganbayeva analyzed the data, prepared figures and/or tables, authored or reviewed drafts of the article, and approved the final draft.
- Asem Turarbek analyzed the data, authored or reviewed drafts of the article, corpora collection for the experiments, and approved the final draft.

## Data Availability

The total corpora of 380 thousand parallel Kazakh–English sentences are available at GitHub and Zenodo:

– GitHub: https://github.com/VladislavKaryukin/kk_en_corpora/tree/v1.

– Zenodo: Vladislav Karyukin, Diana Rakhimova, Aidana Karibayeva, Aliya Turganbayeva, & Asem Turarbek. (2022). VladislavKaryukin/kk_en_corpora: The Kazakh–English parallel corpora (Version v1) [Data set]. Zenodo. https://doi.org/10.5281/zenodo.7115360.

The OpenNMT framework with configurations for the English–Kazakh NMT is available at GitHub and Zenodo:

– GitHub: https://github.com/VladislavKaryukin/OpenNMT_Kazakh-English_NMT/tree/v1.

– Zenodo: Vladislav Karyukin, Diana Rakhimova, Aidana Karibayeva, Aliya Turganbayeva, & Asem Turarbek. (2022). VladislavKaryukin/OpenNMT_Kazakh-English_NMT: The OpenNMT based Kazakh–English NMT (Version v1). Zenodo. https://doi.org/10.5281/zenodo.7118550.

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
