# Peer review of "The neural machine translation models for the low-resource Kazakh–English language pair"

_PeerJ Computer Science, doi:10.7717/peerj-cs.1224_

## Round 0.1 · original submission · Major Revisions

The authors should clearly highlight the achievements of the work. More importantly, they should check for language before submission.

·

Basic reporting

Writing can be significantly improved. Here are some comments that can be used as a guide to improve the overall presentation of the manuscript:

1. Captions of the figures and tables should be self-contained. Readers should be able to understand the illustrations without referring to the main text. Please make sure that the variables in the illustrations are clearly defined.
2. Lines 233-307 discuss the main methodology and not the main results. I suggest most of these be moved to the Materials and Methods section.
3. The summaries of BT and FT approaches can also be improved. Figures 1 and 2 are not consistent with the stated steps. The variables (e.g. T_p, X, and Y) do not appear in the figures. I would like to suggest that the authors use an algorithm environment in Latex to give a better presentation of these steps. This way, the inputs and outputs are clearly stated.
4. The English language should be improved to make so that international readers can fully understand the paper.
5. Please do not start a sentence with a citation (see, e.g., lines 92, 109, 111, 115, and 123). As a suggestion, the authors can instead write "Kandimalla et al. investigated NMT for .... from the OpenNMT tool [9]".
6. The Related work section can be merged into the Introduction section to make the presentation more concise.
7. In the review, please cite other examples of low-resource languages.
8. Some sentences are redundant. As examples, the first sentence in line 172 and the last sentence in line 237 can be omitted. Some parts of the Related works have already been mentioned in the Introduction.
9. Please make sure that references conform with the journal's style. Website links should also be cited as references and not mentioned in the main text. An example of how webpages can be cited can be found in the journal's site: https://peerj.com/about/author-instructions/cs

Experimental design

1. In the corpora formation, only the FT approach was applied. Why mention the other approaches in the Materials and methods section? Did the authors also perform experiment with these other approaches? If yes, additional results should be given.
2. When the metrics (BLEU and TER) are mentioned in the introduction, it is better to state the usual range of values for the scores and define which values are preferable so that the readers can gauge if the values mentioned in the introduction are good or not.
3. Why is the BLEU formula different from the one in the literature? See: Papineni, K., Roukos, S., Ward, T., & Zhu, W. J. (2002, July). Bleu: a method for automatic evaluation of machine translation. In Proceedings of the 40th annual meeting of the Association for Computational Linguistics (pp. 311-318). Moreover, the values of BLEU in [18] and [19] range from 0-100, and not 0-1.
4. The only result of this paper is presented in Table 2. However, the discussion of the results is lacking.
5. Is it fair to compare the results of the study with that of [18] and [19]? Did they use the same dataset? Although the results of this study has better BLEU values, the authors should be transparent with the comparison to make sure that it is not biased.
6. I suggest adding more results. Perhaps adding some sample translations where all the methods work? And translations where only the best architecture was successful?
7. A detailed methodology on corpora formation was given. However, this is not supported by results. I suggest that authors add another comparative study of these different low-resource language approaches.
8. I also suggest that the authors create a flowchart or an algorithm of the best combination that they found. The authors can add more simulations based on this algorithm. You can base these algorithm from lines 329-334.
9. The terms in Table 2 are confusing. The variables "word_tokenized" and "bpe" were not defined properly.

Validity of the findings

1. The data and codes are available. But the text should be modified because it says the source is from GitHub but once you click the link, you will be directed to zenodo. Either change the link to the actual Github links or change the text from Github to zenodo.
2. More results are needed so that the research questions are answered appropriately.
3. Is this approach also applicable to other languages? Application of the proposed approach to other low-resource languages can also be added as future work.
4. Please add limitations of the study.

Additional comments

I find the paper interesting and worthy of publication. However, major revisions should be done first before it can be accepted for publication. I will be very happy to review the revised version of this manuscript. Good luck!

Reviewer 2 ·

Basic reporting

1. The research motivation is not clear and strong enough. The shortcomings of existing studies and the research gap should be summarized.

2. The authors are suggested to add a table for summarizing and defining all the abbreviations.

Experimental design

1. The main contributions are not clear and should be added in the Introduction section.

2. How this research fills an identified knowledge gap is not clear. From my humble knowledge, the most important contribution is the corpora. The models are from previous studies.

3. The data collection and preprocessing steps are not explained comprehensively.

Validity of the findings

N/A

Additional comments

Overall this study is a solid work. However, the research scope is narrow and the research contributions are limited.

Reviewer 3 ·

Basic reporting

This paper creates 380 thousand parallel Kazakh-English sentences and tries to build a high-quality translation model by using a neural network on them. It has a positive effect on improving the machine translation of low-resource languages. However, there are some inadequacies that need to be improved.
The Introduction introduces too little research on related technologies and does not explain the advantages and disadvantages of previous related technologies.
The Related Work lists the past technical research, which should be discussed in categories.
How about the research on English-Chinese translation? Is it a high-resource language? I think there are many Chinese readers of this journal who are interested in this question.

Experimental design

The research questions are clearly defined, relevant and meaningful. However, the following problems should be solved.
First, it is not very clear what is the novelty of the article and which model is finally adopted or proposed by the author.
Second, the model architecture or structure shown in the figures seems somewhat simple, and no new work or contribution to the research field.

Validity of the findings

The experimental data and parameters are described clearly, but the experimental results are simply listed, lacking specific analysis of the experimental results.
The experimental result shown in table 2 is insufficient. More comparisons are needed to show your work is correct and advanced.

Additional comments

no comment

---

## Round 0.2 · Minor Revisions

Please address the concerns raised by Reviewer 3. This is important to ensure the quality of the manuscript.

·

Basic reporting

I commend the authors for addressing my suggestions. The manuscript has improved significantly. However, I still have some minor comments:

1. Abstract is too long and can be more concise.
2. The figures can be improved. For Figures 1 and 2, Tp and D*\cup Y are not written inside a math environment. I suggest that the authors use the same font size and style for the figures and the main text to make them more consistent. The texts in Figures 1 and 2 are also pixelated. In Figure 4, I suggest that texts are not written over the arrow.

Experimental design

No more comments.

Validity of the findings

No more comments.

Reviewer 2 ·

Basic reporting

This version has improved a lot and no further comments.

Experimental design

No comment

Validity of the findings

No comment

Reviewer 3 ·

Basic reporting

no comment

Experimental design

no comment

Validity of the findings

no comment

Additional comments

no comment

---

## Round 0.3 · accepted · Accept

As per the reports of the reviewers, I agree to accept the paper in its current form.

·

Basic reporting

No more comments.

Experimental design

No more comments.

Validity of the findings

No more comments.

Additional comments

The paper has improved significantly from the first version. It can be accepted in its current form.

Reviewer 2 ·

Basic reporting

no comment

Experimental design

no comment

Validity of the findings

no comment

Additional comments

no comment

Reviewer 3 ·

Basic reporting

no comment

Experimental design

no comment

Validity of the findings

no comment

Additional comments

no comment